# Application of Artificial Intelligence in the Management of Pancreatic Cystic Lesions

**DOI:** 10.3390/biomimetics7020079

**Published:** 2022-06-14

**Authors:** Shiva Rangwani, Devarshi R. Ardeshna, Brandon Rodgers, Jared Melnychuk, Ronald Turner, Stacey Culp, Wei-Lun Chao, Somashekar G. Krishna

**Affiliations:** 1Department of Internal Medicine, Ohio State University Wexner Medical Center, Columbus, OH 43210, USA; shiva.rangwani@osumc.edu (S.R.); devarshi.ardeshna@osumc.edu (D.R.A.); 2College of Medicine, The Ohio State University, Columbus, OH 43210, USA; brandon.rodgers@osumc.edu (B.R.); jared.melnychuk@osumc.edu (J.M.); ronald.turner@osumc.edu (R.T.); 3Department of Biomedical Informatics, The Ohio State University College of Medicine, Columbus, OH 43210, USA; stacey.culp@osumc.edu; 4Department of Computer Science and Engineering, The Ohio State University, Columbus, OH 43210, USA; chao.209@osu.edu; 5Department of Gastroenterology, Hepatology, and Nutrition, The Ohio State University Wexner Medical Center, Columbus, OH 43210, USA

**Keywords:** pancreatic cystic lesions, artificial intelligence, radiomics, endoscopic ultrasound, IPMN, genomics

## Abstract

The rate of incidentally detected pancreatic cystic lesions (PCLs) has increased over the past decade and was recently reported at 8%. These lesions pose a unique challenge, as each subtype of PCL carries a different risk of malignant transformation, ranging from 0% (pancreatic pseudocyst) to 34–68% (main duct intraductal papillary mucinous neoplasm). It is imperative to correctly risk-stratify the malignant potential of these lesions in order to provide the correct care course for the patient, ranging from monitoring to surgical intervention. Even with the multiplicity of guidelines (i.e., the American Gastroenterology Association guidelines and Fukuoka/International Consensus guidelines) and multitude of diagnostic information, risk stratification of PCLs falls short. Studies have reported that 25–64% of patients undergoing PCL resection have pancreatic cysts with no malignant potential, and up to 78% of mucin-producing cysts resected harbor no malignant potential on pathological evaluation. Clinicians are now incorporating artificial intelligence technology to aid in the management of these difficult lesions. This review article focuses on advancements in artificial intelligence within digital pathomics, radiomics, and genomics as they apply to the diagnosis and risk stratification of PCLs.

## 1. Introduction

With the reduced cost and increased utilization of diagnostic testing for abdominal pathologies and an overall aging population, the rate of incidentally detected pancreatic cystic lesions (PCLs) has increased compared with previous decades and was recently reported at 8% (95% CI: 4–14%) [1]. PCLs present a unique challenge, as some of these lesions are pre-cancerous, such as mucinous PCLs, cystic neuroendocrine tumors, and solid pseudopapillary tumors. A specific subtype of PCLs, branch duct intraductal papillary mucinous neoplasm (BD-IPMN), is the most common precancerous PCL. BD-IPMNs are increasingly prevalent with increasing age at around 10% [1,2]. At the other end of the spectrum, some of the PCLs—such as serous cystadenomas (SCAs) and pseudocysts—represent benign lesions. Each of these PCLs carries a different risk of malignant transformation, which is outlined in Table 1 [2,3,4,5,6,7,8,9,10,11,12,13,14]. Therefore, it is imperative to correctly identify these cystic lesions and their characteristics to stratify their malignant potential. Classification systems such as the American Gastroenterology Association guidelines (AGA), American College of Gastroenterology guidelines, United European Gastroenterology guidelines (UEG), and Fukuoka/International Consensus guidelines (Fukuoka-ICG) rely on demographics, clinical features, fluid cytology, and cyst morphology based on CT, MRI, and endoscopic ultrasound (EUS) for PCL risk stratification [3,4,15,16].

Patients with PCLs will undergo multiple diagnostic testing modalities, including standard of care (SOC) imaging (CT/MRI) and EUS-guided fine needle aspiration (FNA). Some center may offer novel diagnostics that include cyst fluid next generation sequencing (NGS) analysis, EUS-guided through-the-needle biopsy (EUS-TTNB), and EUS-guided needle-based confocal laser endomicroscopy (EUS-nCLE), thus providing abundant information to clinicians [17]. As there are a multitude of PCLs with varied morphologies ranging from serous to mucinous, clinicians have had difficulty with accurately diagnosing and risk stratifying these lesions. Even with the multiplicity of guidelines and multitude of diagnostic information, risk stratification of PCLs falls short. Studies have reported that 25–64% of patients undergoing PCL resection have pancreatic cysts with no malignant potential, and up to 78% of mucin-producing cysts (IPMNs, mucinous cystic neoplasms (MCNs)) resected harbor no malignant potential on pathological evaluation [17,18,19]. This dissonance between risk stratification using SOC and pathological diagnosis is particularly troublesome in pancreatic surgery, as pancreaticoduodenectomies are associated with a morbidity of 30% and mortality rates of 2.1–5% [20,21]. While the goal of surgery in BD-IPMNs is to resect lesions with advanced neoplasia (high-grade dysplasia or carcinoma), multiple surgical series and a recent international multicenter study showed that 63% of resected BD-IPMNs had low-grade dysplasia (LGD), often representing overtreatment [1,22,23,24,25]. Given the difficulty with accurate current risk stratification and the risks of pancreatic surgery, researchers have utilized novel diagnostic models including computer-aided diagnostics with machine learning (ML) and artificial intelligence (AI) in order to risk-stratify PCLs.

### 1.1. Artificial Intelligence:

Artificial intelligence has applications across multiple practices and has recently been used in medicine to detect and diagnose disease. Given the plethora of diagnostic imaging, genomic information, and endoscopic information in patients being worked up for gastroenterological disease, practitioners have begun to use AI adjunctively to aid in detection and diagnosis of precancerous and cancerous lesions. Since 2010, AI has had multiple applications in gastroenterology, including endoscopic evaluation of lesions, analysis of inflammatory lesions on imaging, and assessment of liver fibrosis [20]. For example, in 2021, the FDA approved an AI assistance tool to help endoscopists identify lesions of concern during colonoscopies. Descriptions of AI terminology can be found in Table 2. AI lies on a spectrum of complexity ranging from machine learning (ML) to deep learning (DL). ML uses mathematical algorithms to analyze input values and predict output values without being explicitly programmed to do so [26]. There are numerous examples of ML, including linear discriminants, Bayesian networks, random forest, and support vector machines (SVMs) [26,27,28]. Each ML method of analysis requires training with data sets that have input data and outcome data [29].

Artificial neural networks (ANNs) are AI computing algorithms that mimic human neurons. Each ANN has an input layer that receives a signal (in the form of data) and an output layer that categorizes the input signal. Just as the neurons weigh action potentials to determine propagation of a signal, hidden layers between the input and output layers of ANNs weigh data characteristics and input from other neurons to determine the output [20,26,29]. DL is comprised of multiple ANNs, allowing for complex processing and automatic detection of the relevant features of input data [30]. DL can be supervised, semi-supervised, or unsupervised. DL algorithms have been applied to numerous specialties within medicine, and convoluted neural networks (CNNs) have recently outperformed both dermatologists and pathologists in identifying pathologies in photographs [31,32]. Advancements in AI have been applied to detection, classification, and diagnosis of PCLs in recent years. Artificial intelligence provides clinicians with modalities to process deeper layers of data on PCLs that are not discernable to humans. The intersection of AI and diagnostic modalities for PCLs, including digital pathomics, radiomics, and genomics, will allow improved risk stratification of these complex lesions. The remainder of this paper focuses on recent updates of AI utilization in PCL evaluation.

### 1.2. The Application of AI to Manage Pancreatic Cystic Lesions: EUS

The increased availability of EUS has allowed for enhanced pancreatic pathology surveillance. Due to the proximity of the transducer to the pancreas, EUS offers more granularity and improved spatial resolution in pancreatic imaging when compared with computed tomography (CT) and magnetic resonance imaging (MRI) and can be used to evaluate lesions as small as 1 cm [33]. The accuracy of CT and MRI in detecting pancreatic lesions < 30 mm is 53% and 67%, respectively, while EUS is 93% accurate [34,35]. Further, novel techniques such as contrast-enhanced EUS, EUS elastography, and EUS-nCLE have improved the diagnostic power of EUS when evaluating PCLs [35]. Given the amount of information in EUS images that is not perceptible to the human eye, clinicians are developing unique ML and DL algorithms to aid in PCL diagnosis and risk stratification.

AI has been utilized in EUS evaluation of pancreatic lesions since 2001, when it was first applied to differentiate focal pancreatitis from malignancy. In this study, a computer algorithm differentiated pancreatitis from malignancy with 73% specificity and 83% accuracy, compared with a blinded endosonographer with 89% and 85%, respectively (*n* = 35) [36]. Early on, AI models mainly employed ML techniques such as linear discrimination, Bayesian theory, and SVM when evaluating EUS images; however, with advancements in AI, DL concepts such as CNNs are now being applied to assist in the diagnosis and stratification of pancreatic lesions in EUS images. Multiple studies show that CNNs and DL models can differentiate between autoimmune pancreatitis, chronic pancreatitis, pancreatic ductal adenocarcinoma, and normal pancreas [37,38,39,40]. Given their varied morphologies and echogenicity, PCLs have been difficult to assess with AI until recently.

## 2. AI and EUS in PCL Risk Stratification

AI serves to help clinicians risk stratify PCLs such as IPMNs and differentiate between types of PCLs by identifying deeper characteristics in EUS images. A 2019 retrospective study (*n* = 50, IPMNs with malignancy = 23) used a DL algorithm to analyze 3790 EUS IPMN images and to evaluate malignant potential in patients having undergone resection with a pathological diagnosis. This study found the sensitivity, specificity, and accuracy of the DL program’s malignant probability to be 95.7%, 96.2%, and 94.0%, respectively. In comparison, the diagnostic accuracy of human interpretation was 56%, and that of the presence of an intracystic mural nodule was 68% [41]. AI has also been able to differentiate between types of PCLs: CNNs have been used to differentiate between EUS morphologies of MCNs (*n* = 60) and SCAs (*n* = 49) with 82.75% accuracy and a 0.88 (95% CI: 0.817–0.930) area under curve score [42].

## 3. AI and EUS-Guided Advanced Diagnostics

EUS-nCLE provides additional information that allows for real-time, high-resolution microscopic imaging of tissue, which facilitates in-vivo histopathology. EUS-nCLE allows practitioners to differentiate between IPMNs and other PCLs [43,44,45]. An image of an IPMN papilla can be seen in Figure 1, highlighting the vascular core and measurements of the epithelial width and density. Certain characteristics of EUS-nCLE allow for the prediction of advanced neoplasia in IPMNs. These characteristics include increased papillary epithelial width and darkness, which reflect cellular stratification and loss of polarity, respectively. These EUS-nCLE imaging characteristics can be seen with corresponding histopathology in Figure 2, which shows low- and high-grade PCL dysplasia. The EUS-nCLE images can be fed into AI algorithms that allow for pixel image processing to stratify based on epithelial dysplasia. In a 2021 study, researchers developed a CNN algorithm that correctly differentiated IPMNs with high-grade dysplasia from those without [46]. In this study, 15,027 EUS-nCLE video frames from 35 consecutive patients with histopathologically proven IPMN (18 with high-grade dysplasia) were used as inputs, with the model yielding higher sensitivity, accuracy, and comparable specificity in diagnosing high-grade dysplasia when compared with international guidelines: sensitivity (model 83.3%, AGA 55.6%, Fukuoka-ICG 55.6%), accuracy (model 85.7%, AGA 68.6%, Fukuoka-ICG 74.3%), and specificity (model 88.2%, AGA 82.4%, Fukuoka-ICG 94.1%) [46]. One of EUS-nCLE’s limitations is lack of widespread adaption. This is likely due to equipment costs (upfront investment), lack of training and exposure during advanced endoscopy fellowships, subtle variations in the advanced endosonography techniques of an nCLE procedure, and the need to learn novel in vivo histopathological imaging.

## 4. The Future of AI and EUS

The future of AI application to digital pathomics and EUS lies in real-time analytics. The aforementioned studies required trained experts to analyze EUS and EUS-nCLE images to identify regions of interest (ROI), in order to parse out high-utility PCL images from EUS image series to feed into DL models. This prevents on-the-fly real-time evaluations of PCLs. However, recent studies have made advancements in this regard. A 2021 study was able to correctly identify and segment PCL ROIs using DL [34]. Further, a recent study using a CNN-based model correctly identified areas of high-grade dysplasia in a series of IPMN EUS-nCLE images, showing concordance with expert validation with an area under the curve (AUC) sensitivity, and specificity of 0.84, 69%, and 89%, respectively [47].

### 4.1. Radiomics

Even with the advent of less-invasive options for the assessment of PCLs such as EUS-FNA and EUS through-the-needle biopsies (EUS-TTNB), there is still a significant morbidity with these procedures of 2.66–10% and 5–10%, respectively [48,49,50,51]. Radiomics, also referred to as quantitative imaging, is an emerging field within radiology that utilizes feature extraction and subsequent machine-based analysis of pixels or voxels from cross-sectional imaging to help create “radiological phenotypes” that can aid physicians in the diagnostic workup [50,52,53,54]. Non-invasive diagnostic options such as radiomics offer the potential to discern between PCLs without concurrent morbidity or mortality risk exposure.

Previous studies utilizing quantitative imaging with data obtained through computed tomography (CT) to discern between subtypes of PCLs have been promising, with reported accuracies of up to 84% [55,56]. When combining these machine-based learning models with the expertise of clinicians, appropriate identification of PCL subtypes reached near perfect [57,58]. However, there are limitations to the capabilities of CT. One example is the identification of isoattenuating lesions, which, for example, can represent up to 10% of pancreatic ductal adenocarcinomas (PDACs) [59]. This has led researchers to explore the capabilities of MRI-based radiomics for the identification and prognostication of PCLs. Udare et al. recently published a comparative analysis between CT- and MRI-based quantitative imaging to distinguish between benign and malignant PCLs that revealed comparable sensitivities and specificities [60].

#### 4.1.1. Radiomics and PDAC

Discerning between PDACs and mass-forming chronic pancreatitis (MFCP) is challenging because of their similar clinical presentations and radiological features; however, these pathologies have drastically different outcomes, thus making an accurate diagnosis important. It has been reported that up to 11% of patients who have undergone a pancreaticoduodenectomy because of concern for PDAC were ultimately diagnosed with benign lesions of the pancreas [61]. Recently, Deng et al. published models discerning between PDAC and MFCP using arterial, portal, T1-weighted imaging (T1WI), and T2-weighted imaging (T2WI) phases of dynamic contrast-enhanced MRI [62]. Notably, these models performed better than a traditional clinical model as well as radiologists’ evaluation of the lesion on MRI. This has since been built upon by Liu et al., who developed a multiparametric MRI radiomics signature model that, when combined with independent clinical risk factors such as carcinoembryonic antigen (CEA) and carbohydrate antigen 19-9 (CA 19-9) levels, demonstrated an AUC of 0.973 (95% CI: 0.904–0.997) [63]. MRI-based radiomic models which focus on tumor morphology have also been used to prognosticate and determine adjuvant therapy response in PDAC [64,65].

#### 4.1.2. Radiomics and PCLs

High-risk PCLs such as IPMNs have also been evaluated with MRI-based radiomic models, which have proven to be superior in predicting malignant potential compared with CT models (AUC = 0.940 vs. 0.864, respectively) and yield improved radiometric feature reproducibility (89.4% for MRI vs. 60.5% and 66.8% in the CT arterial and venous phases, respectively) [66]. IPMNs with a high probability of malignant transformation demonstrated high intratumor entropy, and with optimal threshold entropic values set at >5.27, an accuracy of 83.3% was demonstrated. Similar to PDAC models, the incorporation of clinical factors such as pancreatic duct dilation, CEA, and CA 19-9 levels improved radiomic models in predicting high risk IPMNs (AUC = 0.836 vs. 0.903, respectively) [67,68]. Solid pseudopapillary neoplasms (SPNs) are another high-risk subtype of PCLs that confer a high rate of misdiagnosis, which affects timely treatment strategies and patient outcomes. Radiologically, these lesions appear very similar to other high-risk PCL subtypes such as cystic neuroendocrine tumors, as well as SCAs. To discern between these lesions, Gu et al. created an MRI-based radiomic model utilizing T1WI, T2WI, diffusion-weighted imaging (DWI), and contrast-enhanced (CE) sequencing that was able to achieve an accuracy of 84.1% when combined, which improved with the addition of clinical factors such as CEA and CA 19-9 levels [69].

There are some limitations to MRI-based radiomic models that must be considered. First, most studies have only been single-center analyses, and thus external validity and reproducibility between hospital systems is needed. Second, there is less standardization with MRI as compared with CT, and there is a broader range of scanner properties, image settings and sequences, and feature analyses that exist [70]. However, similar limitations also exist for CT-based radiomic models. To expand this field and improve patient outcomes, the development of standardized protocols and multi-center analyses is paramount.

### 4.2. Genomics

#### 4.2.1. Current Use of Genomics in PCL Management

Genomic analysis is a sparingly used diagnostic technique in PCL management. Currently, the two major applications of this technique are diagnosis of cyst type and malignant risk-stratification. Positive cyst fluid cytology is a surgical indication amongst all current guidelines because it is highly specific for lesions harboring advanced neoplasia, although it is of limited sensitivity [71,72] Genomic analysis via NGS of cyst fluid is a well-studied method that lends to improved sensitivity while maintaining high specificity for the diagnosis of mucinous neoplasm and advanced neoplasia. Specifically, the presence of KRAS and GNAS mutations in cyst fluid has enhanced sensitivity and specificity for classifying mucinous lesions when compared with CEA and cytology [73,74]. Additional mutations (CDKN2A, SMAD4, PTEN, PIK3CA, and TP53) have been identified that are able to predict which lesions are at high risk for progression to malignancy with a sensitivity and specificity of 89% and 100%, respectively [74]. The use of genetics in PCL management has been clearly shown to be of potential benefit when compared with current diagnostic methods. The advancement of artificial intelligence brings the opportunity for improved lesion characterization via AI-based genetic analysis.

#### 4.2.2. AI Genomics in Early Detection of Pancreatic Ductal Adenocarcinoma

Currently, the most utilized biomarker for early detection of PDAC is CA 19-9, but the sensitivity and specificity are suboptimal [75]. One application of AI is the identification of predictive biomarkers for PDAC. For example, one study used an SVM learning algorithm to identify a nine-gene signature that could identify early-stage PDAC with an accuracy, sensitivity, and specificity of 97.53%, 97.96%, and 93.22%, respectively [76]. Furthermore, the PancRISK trial demonstrated AI’s ability to analyze data and create risk-scoring systems. In this trial, three urine biomarkers (LYVE1, REG1B, and TFF1) were used to train and validate five ML algorithms (logistic regression, neural network, random forest, SVM, and neuro-fuzzy technology) in stratifying patients as “Normal” or “Abnormal” risk for PDAC. All algorithms performed similarly (Table 3), but combining PancRISK and CA 19-9 gave a specificity of 0.96 and sensitivity of 0.96 for detecting PDAC, which is better than the current standard of practice [77].

Furthermore, recent studies have utilized mRNA expression, DNA methylation, and miRNA not only as PDAC diagnostic markers but also to appropriately differentiate tumors into less and more aggressive subtypes [78,79,80]. Similarly to PDAC, researchers are applying AI to identify diagnostic and risk-stratification genetic markers for PCLs.

## 5. Risk Stratification Using AI + Genomics in PCL

It is important to accurately stratify low-risk from high-risk PCLs to shield patients with low-risk lesions from the morbidity of pancreatic surgery and conversely appropriately manage and monitor those with high-risk lesions. Through the input of integrated genomic analysis, clinical characteristics, and imaging features, AI has demonstrated the ability to risk-stratify lesions as either low-grade dysplasia or advanced neoplasia. A large multicenter study performed in 2019 by Springer et al. applied a machine learning technique to develop CompCyst. CompCyst utilizes cyst fluid genetic analysis, along with parameters discussed above, to classify patients into one of three groups: surgery, routine monitoring, or suitable for discharge. CompCyst significantly outperformed current clinical management in identifying patients with low risk for malignancy that could be safely discharged (60% vs. 19%, *p* = 1.3 × 10−4) [17]. In the subset of patients without advanced neoplasia (*n* = 140), CompCyst accurately recommended surveillance 68% of the time, whereas the standard of care only recommended monitoring 34% of the time (*p* = 0.02). Overall, use of the AI-based CompCyst test could have prevented unnecessary surgery in 60% of patients in this study. In addition, both standard management and CompCyst similarly identified lesions that required surgery with high accuracy (89% and 91%) [17].

Machine-learning algorithms have been used indirectly to identify markers related to the grade of cyst dysplasia to develop a risk stratifying biosignature. Maker et al. used an SVM-training algorithm to analyze the cyst fluid gene expression of mRNA, miRNA, KRAS, and GNAS. Through the application of AI, they were able to select appropriate genetic targets for the creation of a PCR-based assay that could characterize lesions as low-grade dysplasia or advanced neoplasia. Of all markers that underwent analysis, their AI algorithm selected the mRNA of IL1β, MUC4, and prostaglandin E synthase 2 to be the most accurate for discrimination of high-risk vs. low-risk cysts (AUC 0.86, *p* = 0.002) [81] There is yet to be significant research into AI-based genomics as the sole diagnostic and risk stratification tool. Patient outcomes could be further improved through computer-aided genomic algorithms and optimized biomarker panels.

## 6. The Future: Integrative Computational Models

At present, risk stratification of PCLs based on SOC variables relies on expert consensus-based guidelines, such as Fukuoka-ICG criteria [3,4,15,82]. Multiple studies have addressed the validation of the Fukuoka-ICG criteria and shown a wide range of sensitivity (50–80%) and specificity (55–85%) for the detection of high-grade dysplasia in BD-IPMNs [83,84,85]. While these expert-led consensus guidelines are crucial for the management of PCLs, the non-computational risk assessment of the complex multitude of data may oversimplify underlying relationships in the input risk predictors. Currently in medical practice, clinicians combine available data, apply guidelines such as Fukuoka-ICG, and often confer in multidisciplinary team meetings in order to estimate the risk of malignancy of PCLs. While each of the SOC modalities and advanced diagnostics (EUS-nCLE and cyst fluid NGS) can risk stratify these lesions, an ML/DL-powered integrative computational model can potentially optimize the combined diagnostic accuracy of the data available to clinicians in PCL evaluation (Figure 3). New evaluative modalities that use the tools of AI, such as CompCyst, may allow clinicians to more accurately risk-stratify PCLs and thereby more effectively and efficiently manage these complex lesions.

## Figures and Tables

**Figure 1 biomimetics-07-00079-f001:**
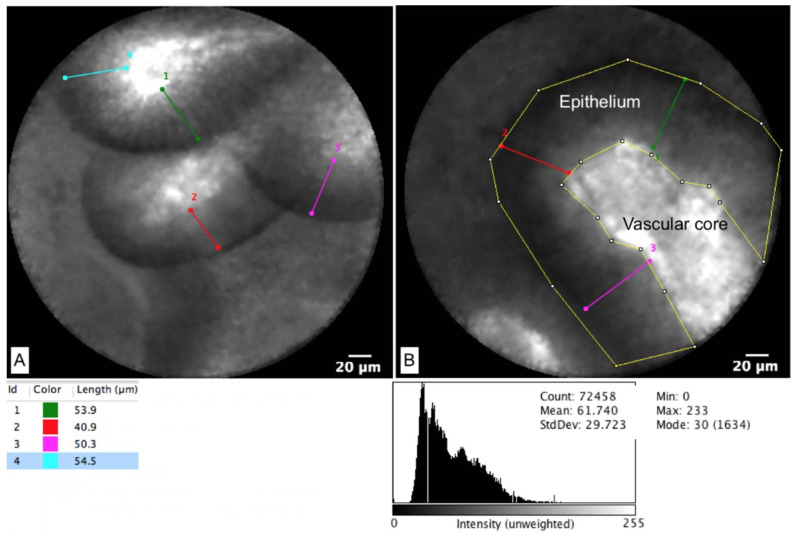
EUS-nCLE image of IPMN. **A: Left panel**: IPMN epithelium and vascular core. Each linear marking corresponds to a different epithelial thickness, as shown in the adjacent measurement. **B: Right panel**: IPMN epithelium and vascular core, with measurements of epithelial density as proxied by pixel intensity in image, with corresponding histogram of mean pixel intensity.

**Figure 2 biomimetics-07-00079-f002:**
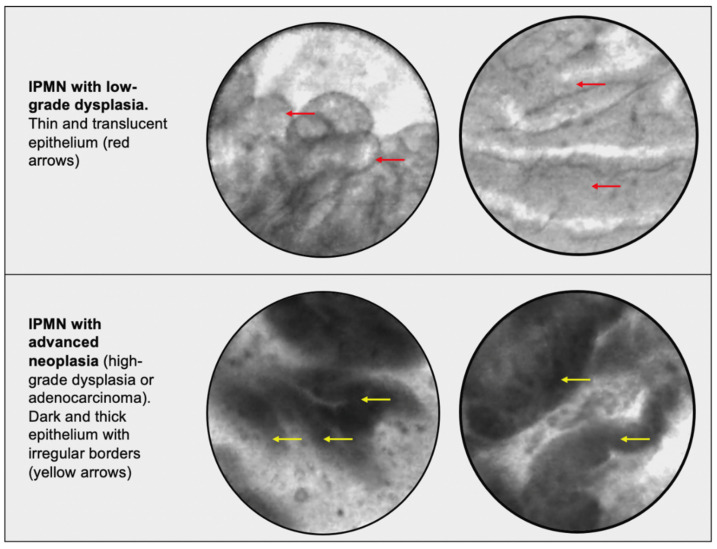
A comparison of EUS-nCLE images. **Top panel**: IPMNs with low grade dysplasia. The thin and translucent epithelium is noted by red arrows on EUS-nCLE images. **Bottom panel**: IPMN with high grade dysplasia. The thicker and darker epithelium is noted by yellow arrows.

**Figure 3 biomimetics-07-00079-f003:**
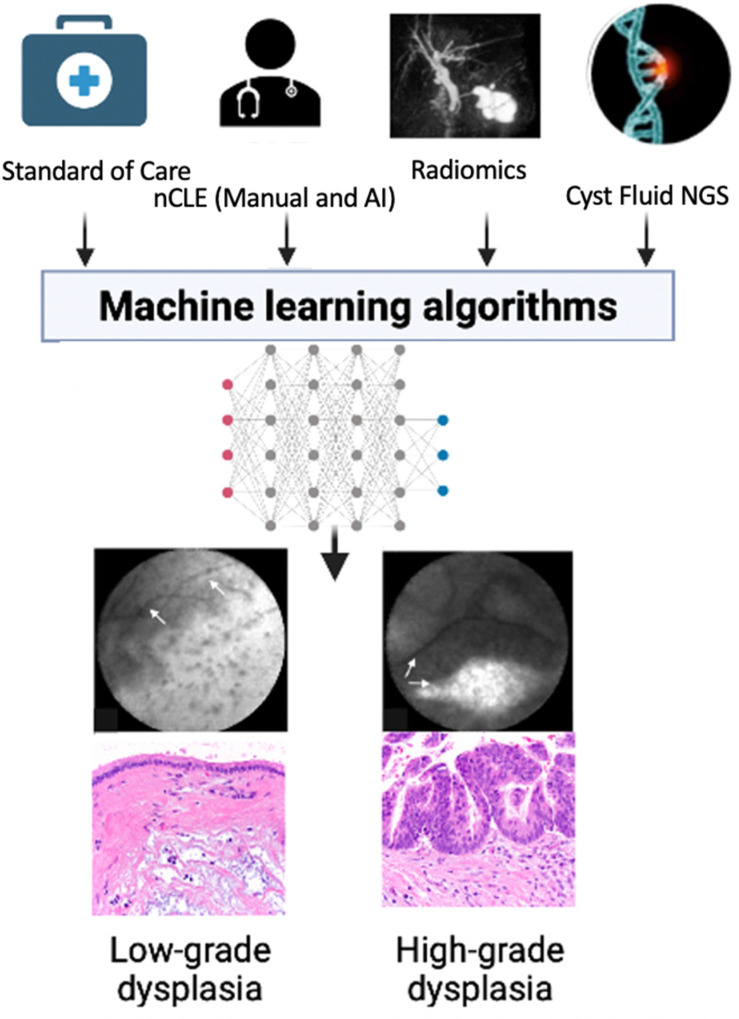
Integration of diagnostics for the prediction of HGD-Ca: Standard of care (SOC) variables: Demographics and patient characteristics; age, gender, onset of diabetes, family history symptoms, pancreatitis history, serum CA 19-9, and cyst fluid analysis (glucose, CEA, cytology). Cyst and pancreas morphology: CT/MRI/EUS: size, wall, thickness, mural nodules, growth rate, and pancreatic duct diameter. nCLE: needle-based confocal laser endomicroscopy. NGS: Next generation sequencing.

**Table 1 biomimetics-07-00079-t001:** Type, characteristics, and malignant potential of pancreatic cystic lesions.

Cyst Type	Characteristics	Rate of Malignancy (%)
Main duct IPMN	Mucinous cyst with variable malignant potential, characterized by main pancreatic duct dilation > 5 mm in the absence of other causes of obstruction [3]	38–68% [2]
Branch duct IPMN	Mucinous cyst with variable malignant potential, characterized as a cyst > 5 mm in diameter that is in communication with the main pancreatic duct. Most common IPMN type [3].	15–17% [3,11]
Mixed IPMN	Displays features of both MD-IPMN and BD-IPMN [3]	28-31% [10]
Mucinous cystic neoplasm	Found almost exclusively in middle-aged women. Mucinous cyst most commonly found in the body or tail of the pancreas. Usually no communication with the pancreatic duct. Columnar epithelium with ovarian stroma differentiates from IPMN [5].	10% [6]
Cystic pancreatic neuroendocrine tumor	Can be solid, cystic, or mixed composition. Can mimic other cyst types on imaging. Can be associated with Multiple Endocrine Neoplasia type 1 (MEN1) [13].	6–31% [14]
Serous cystadenoma	More common in women. Benign, usually found in the tail of the pancreas. Imaging shows microcystic or macrocystic appearance. Central stellate scar is characteristic but not always present. Associated with von Hippel-Lindau disease [7].	0.01% [9]
Solid pseudopapillary neoplasm	More common in younger women, commonly third decade of life. Can occur anywhere in the pancreas. Small tumors are usually solid. Large tumors usually have mixed solid and cystic components. Generally well encapsulated and carry a good prognosis [12].	10% [8]
Pseudocyst	Benign cyst in patients with history of pancreatitis. Typically high lipase and amylase in cyst fluid.	0% [4]

**Table 2 biomimetics-07-00079-t002:** Artificial intelligence (AI) terms.

Term	Definition	Subset of AI
Machine learning (ML)	Models that use historical data (inputs) to categorize and predict outcomes (outputs). Requires human intervention via algorithm training.	ML
Deep learning (DL)	A subfield of ML that uses layered neural networks to automatically record and categorize data outputs without human intervention.	DL
Linear discriminants	A method used to create a linear combination of characteristics that separates/characterizes data into two subsets	ML
Bayesian networks	A probabilistic model that relies on independent/dependent input variables to identify causal probabilities of scenarios	ML
Random forest	A model made up of a large number of decision trees, each producing their own prediction. Predictions are combined to formulate a more accurate prediction of an event occurrence.	ML
Support vector machines (SVM)	Supervised ML algorithm that is capable of performing regression, classification, and outlier prediction	ML
Artificial neural networks (ANN)	Computing algorithms that mimic the human neuron. Each ANN has an input layer and an output layer. Between these layers are hidden layers in which variables are weighted, similar to action potentials.	ML/DL
Convoluted neural networks (CNN)	Type of ANN that allows for unsupervised evaluation of input data, usually in the form of image, speech, or text	DL

**Table 3 biomimetics-07-00079-t003:** PDAC detection performance of five machine learning algorithms in PancRisk trial.

Method	Logistic Regression	Neural Network	Random Forest	Support Vector Machines	Neuro-Fuzzy Technology
Sensitivity	0.81	0.81	0.86	0.82	0.87
Specificity	0.9	0.9	0.82	0.89	0.9

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
