# Peer review of "Application of Artificial Intelligence in the Management of Pancreatic Cystic Lesions"

_biomimetics, 2022, doi:10.3390/biomimetics7020079_

Round 1

Reviewer 1 Report

This is a nice review which focuses on how AI can be applied to difficult areas of radiology.  The addition of pathology and genomics to these problem areas highlight how AI can bring diagnostic specialties together to address difficult clinical problems. Kindly find the detailed comments in the attachment.

Author Response

Submitted as word document

Reviewer 2 Report

This review by Rangwani et al is very interesting an summarizing the use of AI for pancreatic cystic lesions. This work is well written and the literature properly screened. I have only some minor comments:

- overview of cystic lesions and AI terms is very helpful, maybe include some images

- recent UEG guideline on cystic lesions should be mentioned/discussed.

- authors described in detail use of AI in nCLE. However this technique is very advanced and limited to few expert centers. Thus, this expensive technique has several limitations and until now was not incorporated to daily practice. This limitation need to be discussed.

Author Response

Submitted as word document, please see attachment
